# Treatment burden experienced by patients with lung cancer

**Nicole El-Turk**[1,2]*, **Michael S. H. Chou**[1,2], **Natasha C. H. Ting**[1,2], **Afaf Girgis**[2,3], **Shalini K. Vinod**[2,3,4], **Victoria Bray**[3,4], **Claudia C. Dobler**[1,2,5]

**1** Department of Respiratory and Sleep Medicine, Liverpool Hospital, Sydney, NSW, Australia, **2** South Western Sydney Clinical School, University of New South Wales, Sydney, NSW, Australia, **3** Centre for Oncology Education and Research Translation (CONCERT), Ingham Institute for Applied Medical Research, Sydney, NSW, Australia, **4** Cancer Therapy Centre, Liverpool Hospital, Sydney, NSW, Australia, **5** Institute for Evidence-Based Healthcare, Bond University and Gold Coast University Hospital, Gold Coast, QLD, Australia

\* nicole_elturk@hotmail.com

## Abstract

### Introduction

Patients' burden from lung cancer treatment is not well researched, but this understanding can facilitate a patient-centred treatment approach. Current models of treatment burden suggest it is influenced by a patient's perception of their disease and treatment and their capacity to do the work required to treat their disease.

### Methods

Sixteen patients and 1 carer who were undergoing or had completed conventional or stereo-tactic ablative radiotherapy, chemotherapy or immunotherapy for lung cancer in the last 6 months participated in a semi-structured interview. A treatment burden framework was used with three main themes: a) treatment work, b) consequences of treatment and c) psychosocial factors affecting treatment burden.

### Results

The majority of patients did not feel unduly burdened by treatment tasks, despite having a large treatment-associated workload. Many saw treatment as a priority, causing them to restructure their life to accommodate for it. Patients wished that they would have been better informed about the lifestyle changes that they would have to make before treatment for lung cancer commenced and that the health service would provide services to assist them with this task.

### Discussion

While there was a large burden associated with lung cancer treatment, patients felt motivated and equipped to manage the workload because the disease was considered severe and life-threatening, and the treatment was seen as beneficial. Before initiating treatment for lung cancer, patients should be informed about lifestyle changes they likely have to make and should be offered assistance.

**Data Availability Statement:** All data are freely available and are within the paper.

**Funding:** C.C.D. was supported by an award from the South Western Sydney Local Health District (SWSLHD, https://www.swslhd.health.nsw.gov.au/

) and a fellowship of the Australian National Health and Medical Research Council (grant# APP1123733, https://www.nhmrc.gov.au/). A.G. is funded through Cancer Institute NSW grants (https://www.cancer.nsw.gov.au/). The funders had no role in study design, data collection, data analysis, data interpretation, or writing of the manuscript.

**Competing interests:** The authors have declared that no competing interests exist.

## Introduction

There were an estimated 2.1 million patients diagnosed with lung cancer worldwide in 2018 [1]. With the psychosocial burden experienced from lung cancer already being high [2], it is important that we understand the work that patients must do to undergo treatment, also known as "treatment burden" [3]. While "disease burden" is a well-known epidemiological concept, the concept of "treatment burden" has only emerged in more recent times in the medical literature. Treatment burden describes the work that patients need to do to manage their health and the impact that this workload might have on patients' life [3]. Treatment work includes medication taking, attending medical appointments, monitoring health, diet, exercise and other activities. This workload can impact on patients' social and professional life, finances and emotional state and can be the cause for non-adherence to prescribed treatments.

The cumulative complexity model postulates that there is a balance between a patient's treatment workload and the patient's capacity to deal with the workload [4]. Lung cancer treatment workload includes attending appointments or undergoing therapy, taking medications to manage symptoms and managing treatment side effects [5]. Capacity reflects a patient's ability to complete the required medical treatment and involves their physical and psychological circumstances as well as social context, all of which inform their perception of the treatment workload [6]. If this workload exceeds a patient's capacity, this might result in non-adherence, which negatively affects patient health and quality of life [3]. Assessing a patient's capacity to undergo treatment and navigate the complexity of the healthcare system within their personal context is important in addressing burden of treatment and ensuring that the treatment plan is manageable for the patient [7,8].

Lung cancer treatment varies according to pathology stage and the patient's fitness for various treatments. Currently, there are three main treatment modalities; surgery, radiotherapy, and systemic therapies (including chemotherapy, molecular targeted therapies and immunotherapy). Each modality is associated with a different treatment duration and side effect profile, all of which may impact the perceived burden from treatment [9].

There are a number of studies on treatment burden in other chronic conditions and studies that focused on a single aspect of treatment burden in lung cancer, for example the financial cost or adverse effects of treatment. A cohort study evaluated treatment burden defined as the number of encounters, physicians involved, and medications prescribed in patients with non-small-cell lung cancer [10]. In another study, three patients with non-small cell lung cancer were interviewed about how they experience everyday life during curative radiotherapy [11]. There is also a systematic review of the qualitative studies that explored the experience of patients with lung cancer or chronic obstructive pulmonary disease and/or their informal caregivers with health or social care interactions [12]. However, until now, there has been no study that systematically explored all aspects of treatment burden experienced by patients with lung cancer. The aims of our study were to describe the subjective workload of patients undergoing lung cancer treatment and the impact of this workload on patients' life, and to identify aspects perceived as burdensome by patients as well as potential solutions.

## Methods

This was a qualitative study using semi-structured interviews with patients undergoing treatment for lung cancer at a tertiary hospital's cancer therapy centre in Sydney, Australia. Ethics approval was provided by the South West Sydney Local Health District Human Research Ethics Committee (HE15/304). Written informed consent was obtained for all in person interviews, with approval to obtain oral consent for phone interviews. This was documented on the consent form.

### Recruitment

Interviews were conducted with i) patients who could communicate in English, were over the age of 18 years and were either undergoing treatment or had completed treatment with chemotherapy, immunotherapy, or radiotherapy (either stereotactic ablative body radiotherapy (SABR) or conventional radiotherapy) for lung cancer in the 6 months preceding the interview, or ii) adult (aged 18 years and over) carers of a patient meeting the criteria listed above.

Purposive sampling was used to recruit patients undergoing different types of treatment (chemotherapy, immunotherapy, SABR or conventional radiotherapy) with different treatment intents (curative, palliative). Participants were identified from clinic lists and by searching multi-disciplinary meeting agendas between July 2018 and July 2019. Potential participants were contacted by one of the investigators (NE), informed about the study and invited to participate in an interview, in person or over the phone.

Interviews were conducted one-on-one or with a partner/carer present either in person or over the phone with no time limit and were recorded and transcribed by the interviewer (NE). The interview guide (S1 File) was derived from a preliminary framework of treatment burden in lung cancer, generated from a taxonomy of the burden of treatment [5] and a systematic review of patient-reported measures of burden of treatment in three chronic diseases [7]. It addressed three primary themes: i) treatment work, which included work undertaken that was at the request of a health professional or associated work that was necessary to complete instructions, ii) consequences of treatment (e.g. side effects, lifestyle changes) and iii) psychosocial factors that affect treatment burden. Minor additions were made after the first two interviews to ensure that the topics discussed were addressed in all interviews.

### Data analysis

Interviews were analysed using NVivo 12 qualitative analysis software. Open coding was initially used, with concepts identified in the transcripts being coded into new subthemes under the three themes taken from the preliminary framework of treatment burden in lung cancer. An iterative process was used to code data into the subthemes using both inductive and deductive reasoning [13].

Data collection ceased after data saturation was achieved [14]. Four interviews were co-coded and discussed by two independent researchers (MC, NT) for quality assurance. After analysis of all interview data, the preliminary lung cancer treatment burden framework was updated. The final version (Fig 1) includes all treatment burden themes that were identified as relevant to lung cancer patients.

## Results

### Participants

Thirty-five patients were invited of whom 16 patients and 1 carer participated in the study. Reasons for non-participation included: insufficient English communication skills (n = 4), being too unwell (n = 4), being too busy/uninterested (n = 6). Four patients could no longer be contacted following invitation to participate and one patient passed away before being interviewed.

Participant demographic details are summarised in Table 1. Of two participants who continued working during lung cancer treatment, one was the carer of a patient. Of 8 retired participants, only one participant was already retired when diagnosed, the other seven retired to undergo treatment for lung cancer. The majority of the interviews (n = 14) were conducted in person.

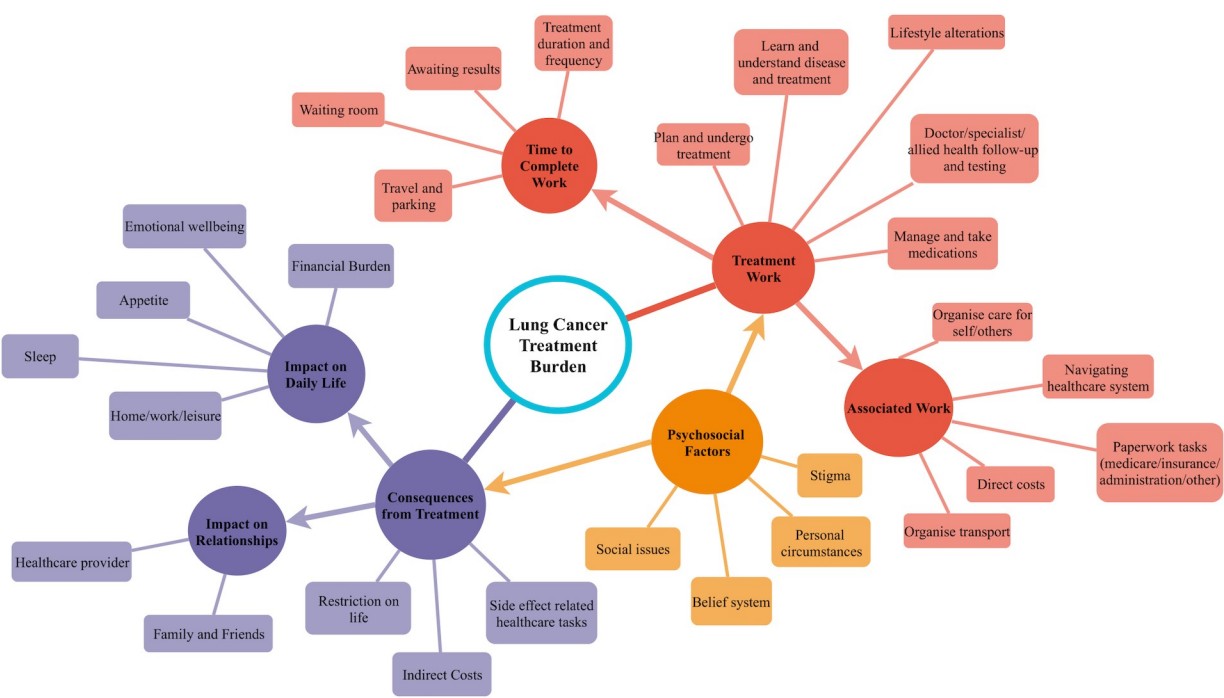

**Fig 1. Lung cancer treatment burden framework.** A diagrammatic representation of aspects of lung cancer that may be considered burdensome.

## Treatment work

Participants put a high value on prolonging survival and were willing to prioritise potentially beneficial treatment for a disease perceived as life-threatening despite the workload. Most patients (n = 13) felt well equipped to complete healthcare tasks including undergoing treatment, making medical decisions, going to appointments and implementing lifestyle changes.

> "*I thought if I had to do [radiation] again, I'm not going to do it, but if it's going to keep me alive a little bit longer, then yeah, I'll do it*"–participant 12

Treatment gave patients hope. Some appreciated life more than before their cancer diagnoses and were motivated to become a better person.

**Treatment.** Patients treated with SABR (n = 5) had less workload than patients receiving other treatments. Chemotherapy, and especially combined chemo- and radiotherapy, were associated with a substantially greater workload. The workload for immunotherapy (infusions every second week for 1 month to 4 years, depending protocol) was rated low by patients.

> "*Chemotherapy is very bad. I'm very sick emotionally, physically. . . This immunotherapy is working very well. I'm good, no side effects.*"–participant 10

Making decisions about proposed treatments was straightforward for patients in this cohort. The desire to undergo treatment was mainly driven by a treatment goal (seeking cure, prolonging life) but also by other motivations (undergoing treatment for the patient's family's sake). Patients relied on being guided in any medical decisions by their doctors' advice and did not consciously weigh benefits and downsides of treatments (including treatment burden).

**Table 1. Participant demographics.**

| Participants | n = 17 | % |
|---|---|---|
| **Age** | | |
| Mean | 67.7 years | |
| Range | 58–80 years | |
| **Interview length** | | |
| Mean | 48.2 minutes | |
| Range | 25–80 minutes | |
| **Interview type** | | |
| In person | 14 | |
| Phone | 3 | |
| **Sex** | | |
| Female | 11 | 64.7 |
| Male | 6 | 35.3 |
| **Relationship status** | | |
| Married/partner | 13 | 76.5 |
| Separated/divorced | 2 | 11.8 |
| Widowed | 1 | 5.9 |
| Never Married | 1 | 5.9 |
| **Occupation status** | | |
| Retired | 8 | 47.1 |
| Homemaker | 6 | 35.3 |
| Unemployed | 1 | 5.9 |
| Currently Working | 2 | 11.8 |
| **Cultural background** | | |
| White | 11 | 64.7 |
| English as a Second Language | 6 | 35.3 |
| **Smoking status**[*] | | |
| Current smoker | 7 | 41.2 |
| Ex-smoker | 7 | 41.2 |
| Never smoked | 2 | 11.8 |
| **Diagnosis**[*] | | |
| Stage I lung cancer | 6 | 35.5 |
| Stage III lung cancer | 5 | 29.4 |
| Stage IV lung cancer | 5 | 29.4 |
| **Intent** | | |
| Curative | 10 | 58.8 |
| Palliative | 7 | 41.2 |
| **Treatment** | | |
| SABR | 5 | 29.4 |
| Radiation therapy | 2 | 11.8 |
| Chemotherapy | 3 | 17.6 |
| Immunotherapy | 7 | 41.2 |
| **Time since treatment** | | |
| Current | 9 | 52.9 |
| <3 months | 5 | 29.4 |
| 3–6 months | 3 | 17.6 |
| **Co-morbid chronic conditions**[*] | | |
| 0–2 | 4 | 23.5 |

(*Continued*)

**Table 1.** (Continued)

| Participants | n = 17 | % |
|---|---|---|
| >2 | 12 | 70.6 |

*excludes carer.

"*[To doctor] I'm in your hands, just do what you think*"–participant 12

One participant highlighted that they were not informed of the impact treatment would have on their life.

Having appointments rescheduled or cancelled without adequate notification was perceived as burdensome. Some patients felt that their appointments with allied health staff were unnecessary or unhelpful. Patients experienced stress and anxieties associated with uncertainty about computed tomography scan results and fear that the cancer was progressing. A participant was burdened by having to repeat scans that already been done externally.

**Lifestyle changes.** Patients were commonly instructed by health professionals to maintain their normal lifestyle. A participant was specifically told to learn to manage their disease and actively incorporate it into their daily life.

"*I was encouraged by doctors before I started chemotherapy. . . try and maintain my life as normal as I possibly can*"–participant 9

Quitting smoking was a struggle and stressful for patients who smoked and were asked to quit. While weight loss/gain was occasionally suggested by doctors, patients did not feel pressured by medical staff to change their diet. Pressure to eat was self-imposed by some participants who recognised that they needed a healthy diet to be sufficiently energised during treatment. Most participants were told to maintain current dietary and exercise habits if they were able to and adjust as necessary.

"*One of the nurses said. . . let your body guide you. If you feel tired, then rest. If you don't feel tired, then go and do something*"–participant 13

**Associated work.** Topics addressed included providing or receiving care (if the interviewee was a carer or patient respectively), interacting with the healthcare system, travelling to appointments and dedicating time to treatment and side effects.

Care was often provided to patients by family and friends and occasionally by healthcare services. Patients often required assistance with household tasks that they were too fatigued to complete, such as cleaning and cooking. Participants who experienced breathlessness required help with showering and dressing. While the assistance was appreciated, the dependence on others was occasionally difficult for patients. During treatment, patients with carer's responsibilities found it challenging to continue caring for others. Two patients delegated care for an elderly family member while undergoing treatment.

"*And from that time when I took my mum there [nursing home] and it was just myself with my husband, he is much calmer, I am much calmer.*"–participant 17

The shortest treatment, SABR treatment was second daily for three to five sessions. On the other hand immunotherapy was given second weekly for 1 month to 4 years. Patients found it

very achievable to dedicate the required time to both these treatments. Conventional radiation therapy demanded more time and effort because it was administered daily, five days a week for four to six weeks. Chemotherapy was delivered either weekly or less frequently. Patients receiving radiation found travelling to hospital appointments especially cumbersome as they spent more time travelling than receiving treatment. Between treatment, travel and managing side effects, the whole day was lost and occasionally the next day, if chemotherapy was administered (up to 3 times a week).

> "*That is the biggest burden, consumption of my time. It takes me away from other activities and planning anything in my life anymore. It's become all about the hospital*"–participant 9

Participants felt that they lost time from their life to treatment, but it was still considered a priority to focus their effort on.

> "*If I didn't have [treatment] I wouldn't be alive*"–participant 16

Travel burden was associated with living far from the hospital, experiencing peak hour traffic, relying on others for transport (especially if patients were instructed not to drive) or having to use public transport. Parking around the hospital was expensive and difficult to find.

## Consequences of treatment

Consequences of treatment included side effects from treatment and the impact that the treatment workload had on patients' psychosocial wellbeing, specifically lifestyle, finances, relationships and emotional wellbeing.

**Side effects.**   Fatigue was the most common side effect of all treatments, had the greatest impact (reducing the ability to complete daily tasks) and usually continued after treatment was ceased.

> "*I just couldn't get out of bed, I was so fatigued. Even now the fatigue has set in again*"–participant 12

Other symptoms included nausea/vomiting associated with chemotherapy, pain and rash with radiation/SABR and myasthenia gravis and pneumonitis with immunotherapy. Participants were well informed about side effects and when to expect them, which helped them cope with them. Side effects occasionally warranted hospital admission.

Participants commonly experienced sleeplessness at night, which they attributed to stress related to their diagnosis, or to an adverse effect of medications, specifically steroids.

**Psychosocial impact.**   Despite being instructed by healthcare workers to 'maintain normal life,' participants felt the need to restructure their lifestyle by reducing their daily workload to accommodate for treatment and side effects.

> "*Stuff like this [lung cancer and treatment] very much forces you to reorganise your entire life*"- participant 10

Fatigue caused patients to abandon household duties during and after treatment, which would either be completed by family members or a private service organised by the patient, or the tasks would not be done, which impacted patients' mental health. Other changes included an increased consumption of takeaway foods, no time or desire to socialise and constraints on patients' ability to work—leading to early retirement. Those who retired

experienced a financial burden from the decreased income, forcing them to restructure their finances and lifestyle.

> "*I've always worked and always been the provider, and I can't do that anymore. So financially we're suffering terribly.*"–participant 9

Of the two participants who continued working, one was in the process of retiring and the other reduced work commitments for treatment.

Participants were relieved they were not required to pay for treatment due to Australia's universal health care system. Those who quit smoking benefitted from having extra money they would have otherwise spent on cigarettes.

Well-informed, trustworthy and helpful nursing/healthcare staff at the treatment center was identified as a factor relieving treatment burden.

Participants described having family/friends take on duties as carers and homemakers. While some participants felt that personal relationships improved while they were undergoing treatment for lung cancer, others felt they were burdening their loved ones.

> "*It upsets me more, because I know I'm putting stress on them [family]. And I don't want to do that*"–participant 13

Treatment occasionally reduced the emotional burden of being diagnosed with lung cancer because it was perceived as a potential cure. Obtaining information about treatment and prognosis reduced anxiety.

> "*The treatment eases the emotional burden a little bit*"–participant 11

Treatment and side effects occasionally had a negative emotional impact. Changes in physical appearance, such as hair loss, or an inability to quit smoking negatively affected patients' confidence. Some participants were frustrated with having to change their lifestyle for treatment. Participants were fearful about the effectiveness of their treatment or the prognosis, especially when they were not eligible for surgery or had treatment withdrawn.

## External factors

Participants described external factors such as co-morbidities, support, social circumstances and stigma that alleviated or worsened their treatment burden.

Management of co-morbidities increased contact with healthcare services. Co-morbidities also impacted lung cancer treatment, e.g. cancer being inoperable due to an existing lung condition and insufficient lung function to withstand removing part of the lung, or were exacerbated by treatment, e.g. radiation treatment worsened shortness of breath in somebody with emphysema.

Having family and friends to help at home or provide emotional support was valued by many patients. Participants accessed hospital services, the Cancer Council (an Australian charity that supports patients diagnosed with cancer) and Facebook groups for assistance. Some also described being supported by members of their religious community and gaining strength from their faith.

Some participants experienced pressure from family to complete treatment and make lifestyle changes, which could exacerbate treatment burden when it did not align with the patient's wishes or opposed the advice provided by health professionals.

"*But the expectations that my family and friends have is that, I don't have a choice*: *'You're doing this and you're going to get better'.*"–*participant 16*

Some social circumstances complicated treatment and increased burden (e.g. a sick spouse or parent). Personal situations were occasionally an incentive to complete treatment, e.g. a young child in the patient's care. Previous exposure to advanced cancer taking the life of family/friends left participants with preconceptions about cancer having poor treatment success rates. Cultural beliefs occasionally influenced patients' outlook and decisions.

Most participants thought that there is a stigma associated with having lung cancer, as the public often sees lung cancer as a self-inflicted disease in smokers. Some patients described feeling stigmatised, even though people around them would usually not say something to that effect directly.

"*People didn't actually say it to your face, but you know what they're thinking*"–*participant 7*

One patient in this cohort was explicitly reprobated for still smoking with a lung cancer diagnosis. Patients who blamed themselves for having caused their cancer by smoking did not feel stigmatised by their environment.

## Solutions to reducing treatment burden

Participants identified areas for improvement to relieve their current treatment burden. Some patients thought that the healthcare system was inflexible and did not respond to their requests. Suggestions for improvement included the ability to choose or vary appointment schedules, providing options to purchase better quality food (especially in the hospital ward) and making information available about services for transport to and from the hospital and community care. There was also a request for doctors to emphasise the importance to quit smoking, with one patient explaining that they have not stopped smoking because they don't feel any pressure to quit.

## Discussion

This is the first original study that explores all areas of treatment burden in lung cancer, i.e. the work that patients need to do to manage their condition and the impact that this workload might have on patients' life, It demonstrated that patients undergoing treatment for lung cancer experience significant treatment burden, especially with regard to the work of completing treatment and the impact of treatment and side effects on lifestyle and relationships. Many participants described having to restructure their life to accommodate treatment but felt motivated and equipped to do so.

Patients who participated in the study might have had better capacity to deal with the challenges of lung cancer treatment than the average patient, therefore the true impact of the workload associated with cancer treatment might be higher than in our patient sample.

This study was conducted with patients treated at a single cancer centre. While some results concerning treatment burden may only represent this population (e.g. cost, as treatment was available for free at this Australian public hospital), most aspects of treatment burden in lung cancer are generic and would likely apply to lung cancer patients in other settings as well. Patients who could not communicate in English without the help of an interpreter were excluded in our study and treatment burden associated with non-English speaking patients was therefore not addressed.

Data saturation was already reached after 16 interviews, possibly because the interview guide was comprehensively addressing aspects of treatment burden, being based on a taxonomy of the burden of treatment and a systematic review of patient-reported measures of burden of treatment in three chronic diseases. The themes covered in the interview with one carer were comparable to the themes discussed by patients. While it would have been desirable to include more carers as participants, our efforts to recruit additional carers as participants during the study period was unsuccessful.

A systematic review about patient capacity showed that reframing one's biography in chronic conditions is important in maintaining quality of life and patients who fail to do so suffer psychologically [15]. Reframing one's biography involves accepting a disease as a component of one's identity and adapting one's lifestyle to accommodate treatment. In our study, patients who experienced less emotional burden accepted the lung cancer diagnosis as part of their identity, which resulted in an easier acceptance of lifestyle changes.

Treatment burden in lung cancer may be readily accepted by patients because of the perception that cancer is a life-threatening disease [16]. This idea is supported by a qualitative study of patients undergoing curative radiotherapy for lung cancer, which identified treatment as being a priority for which they are willing to alter their lifestyle to accommodate for [17]. In our study, patients were willing to accept a large workload because they perceived treatment to be beneficial.

Participants in our study generally accepted the treatment recommended to them by their doctors without asking about the balance of benefits and downsides or asking about alternatives (including doing nothing). A qualitative study about decision-making in second line treatment for cancer found that the decision-making process about cancer treatment is often driven by health professionals instead of being a shared decision-making process, including the patient's values and preferences [18]. Shared decision making in this situation is desirable, as doctors are disease experts but patients are experts in knowing their own values and preferences.

The study participants described not considering burden of treatment in the decision-making process about the preferred treatment for lung cancer. Participants were educated about the disease, treatment and expected side effects, but not the impact of treatment on their life. A German study surveying 338 cancer patients found that patients were dissatisfied with the education concerning management of side effects, suggesting they would have benefitted from advice on strategies to deal with them [19]. While this study was not specific to lung cancer, it highlights that patients value realistic advice on dealing with their disease. Patients in our cohort were given the advice to maintain normal life, which was not realistic in most cases. Providing information about necessary changes in lifestyle may enable patients to more easily transition to a life with treatment.

The capacity-burden model proposes that undue treatment burden is experienced when workload exceeds a patient's capacity [6]. Treatment burden is also experienced when treatment is perceived negatively by a patient [3]. Patients in our study generally had a positive view of treatment as the means that would prolong their life or cure them from cancer. This may explain why most patients in our study did not feel excessively burdened, even when they had to deal with a large treatment workload and had to re-arrange their every-day life to accommodate treatment.

Clinicians may be able to use insights from our study to help ease patients' burden from treatment by encouraging them to proactively make lifestyle changes and providing access to resources, such as home care, transport and respite early on. Clinicians may also advise patients to make treatment their only goal, and to view their usual daily activities as supplementary tasks to complete if they are able. This essentially prepares patients to restructure their lives around treatment, instead of fitting treatment into their schedules.

In summary, this study showed that patients undergoing treatment for lung cancer experienced a considerable treatment burden. They had to re-organize their lives to accommodate treatment, which often (permanently) impacted employment. Patients felt motivated and well equipped to tackle treatment, which was reflected in limited negative emotional impact of the treatment workload. Suggestions to reduce treatment burden include managing patients' expectations, especially with regard to lifestyle changes they will need to make to cope with the demands and consequences of lung cancer treatment, and providing services to assist them. Clinicians should advise patients to proactively restructure their lives around treatment, instead of fitting treatment into their schedule.

## Supporting information

**S1 File. Interview guide.**
(DOCX)

## Author Contributions

**Conceptualization:** Nicole El-Turk, Michael S. H. Chou, Natasha C. H. Ting, Afaf Girgis, Shalini K. Vinod, Claudia C. Dobler.

**Data curation:** Nicole El-Turk, Claudia C. Dobler.

**Formal analysis:** Nicole El-Turk, Claudia C. Dobler.

**Investigation:** Nicole El-Turk.

**Methodology:** Nicole El-Turk, Michael S. H. Chou, Natasha C. H. Ting, Claudia C. Dobler.

**Project administration:** Nicole El-Turk.

**Resources:** Nicole El-Turk, Shalini K. Vinod, Victoria Bray.

**Supervision:** Afaf Girgis, Shalini K. Vinod, Claudia C. Dobler.

**Validation:** Michael S. H. Chou, Natasha C. H. Ting, Claudia C. Dobler.

**Visualization:** Nicole El-Turk.

**Writing – original draft:** Nicole El-Turk, Michael S. H. Chou, Natasha C. H. Ting, Claudia C. Dobler.

**Writing – review & editing:** Nicole El-Turk, Michael S. H. Chou, Natasha C. H. Ting, Afaf Girgis, Shalini K. Vinod, Victoria Bray, Claudia C. Dobler.

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
