## [Decision Letter · Decision Letter 0]

6 Nov 2020

PONE-D-20-16636

Treatment burden experienced by patients with lung cancer

PLOS ONE

Dear Dr. El6turk,

Thank you for submitting your manuscript to PLOS ONE. After careful consideration, we feel that it has merit but does not fully meet PLOS ONE’s publication criteria as it currently stands. Therefore, we invite you to submit a revised version of the manuscript that addresses the points raised during the review process.

As you will see through our two reviewers comments below, several major issues have been raised. However, due to the paucity of data in this field, I am happy to offer you this opportunity to resubmit. In the revised version, would you be so kind to give more space to what's new from previous studies ?

We look forward to receiving your revised manuscript.

Kind regards,

Christophe Leroyer

Academic Editor

PLOS ONE

**Comments to the Author**

1. Is the manuscript technically sound, and do the data support the conclusions?

Reviewer #1: Partly

Reviewer #2: Partly

2. Has the statistical analysis been performed appropriately and rigorously? 

Reviewer #1: N/A

Reviewer #2: I Don't Know

3. Have the authors made all data underlying the findings in their manuscript fully available?

Reviewer #1: No

Reviewer #2: Yes

4. Is the manuscript presented in an intelligible fashion and written in standard English?

Reviewer #1: Yes

Reviewer #2: Yes

5. Review Comments to the Author

Reviewer #1: The text is about the patient’s perceptions of the burden from lung cancer treatment. It is a very interesting topic and a field not very well documented. The text is clear and written with a very good level of language.

The abstract is clear and well built.

The introduction describes the impact of the cancer treatment in the life of the patients and also, the burden of the disease. But the problematic of the research is vague: the authors approach at the line 51 the “additional work of the patients”, but they do not explain the concept: Are they talking about concept of professional didactic as defined by different researches from humanities? In the same way, they use the concept of “burden” but they do not define it. Moreover, they talk about the concept of perception in the abstract, which is one of main concept in psychology, but we do not find it in the introduction? They authors should be more clear about the conceptual basement of this study.

The method refers to a qualitative study witch perfectly adequate with the study of the patient's perception about the life with a disease. The patient recruitment is very clear and well detailed. The corpus constitution and the analysis of the corpus seem rigorous. Unfortunately the authors do not justify the size the sample. Why they have chosen to interview 16 patients and 1 carer participate? Usually, we are not used to pre-defining the size of the sample in qualitative study and to pursue the interviews until the saturation of the information. The experience have shown that the saturation is frequently obtain around 25/30 interviews. Finally, the authors do not explain clearly the guide of interview. It is more helpful to have a classical interview guide whit three or four questions instead of a semantic map.

Concerning the results, it is showed some idea very interesting and the verbatim are really helpful.

The discussion summarises clearly the main results of the study. But, there is lack of analyse with the conceptual framework of the study. Moreover, some elements in connection with the field of health psychology could be seen in the results, in particular ideas relating to the concept of coping or social support that would have been relevant to analyse and put into perspective with this scientific literature

To conclude, it is an interesting text and a good exploratory study in this topic but characterised by a cruel lack of data. To me, the article is not proposed in the appropriate journal due to the methodological weaknesses.

Reviewer #2: This original submission does not provide new or relevant data compared to the literature.

Exclusion criteria are not clear: 2 patients still undergoing treatment were excluded (line 133) but patients undergoing treatment could be included (line 94)

Figure 1 is illegible

6. PLOS authors have the option to publish the peer review history of their article (what does this mean?). If published, this will include your full peer review and any attached files.

Reviewer #1: No

Reviewer #2: No

---

## [Author Response · Author response to Decision Letter 0]

17 Dec 2020

Reviewer’s Comments:

Reviewer #1: 

1. The text is about the patient’s perceptions of the burden from lung cancer treatment. It is a very interesting topic and a field not very well documented. The text is clear and written with a very good level of language. The abstract is clear and well built.

Response: We thank the reviewer for the kind feedback.

2. The introduction describes the impact of the cancer treatment in the life of the patients and also, the burden of the disease. But the problematic of the research is vague: the authors approach at the line 51 the “additional work of the patients”, but they do not explain the concept: Are they talking about concept of professional didactic as defined by different researches from humanities? In the same way, they use the concept of “burden” but they do not define it. Moreover, they talk about the concept of perception in the abstract, which is one of main concept in psychology, but we do not find it in the introduction? They authors should be more clear about the conceptual basement of this study.

Response: The term “additional work” may have been confusing and has now been replaced with “work”. We have specified that this work is also referred to as treatment burden. The concept of treatment work or treatment burden originated from the medical literature (not from the humanities or from the field of psychology). The concept has now been explained in the introduction. We have added the following paragraph:

“While “disease burden” is a well-known epidemiological concept, the concept of “treatment burden” has only emerged in more recent times in the medical literature. Treatment burden describes the work that patients need to do to manage their health and the impact that this workload might have on patients’ life (3). Treatment work includes medication taking, attending medical appointments, monitoring health, diet, exercise and other activities. This workload can impact on patients’ social and professional life, finances and emotional state and can be the cause for non-adherence to prescribed treatments.”

The word “perception” was used to reflect the patient perspective. As this is a medical paper, targeting a clinical readership, we have used the term as it is commonly used in medicine. We acknowledge that in a psychological paper the term might have different connotations.

3. The method refers to a qualitative study witch perfectly adequate with the study of the patient's perception about the life with a disease. The patient recruitment is very clear and well detailed. The corpus constitution and the analysis of the corpus seem rigorous. 

Response: We thank the reviewer for the kind feedback.

4. Unfortunately the authors do not justify the size the sample. Why they have chosen to interview 16 patients and 1 carer participate? Usually, we are not used to pre-defining the size of the sample in qualitative study and to pursue the interviews until the saturation of the information. The experience have shown that the saturation is frequently obtain around 25/30 interviews. 

Response: We did not define a sample size a priori but continued interviewing participants until data saturation was reached, i.e. no new themes were emerging. Data saturation was already reached after 16 interviews, possibly because the interview guide was comprehensively addressing aspects of treatment burden, being based on a taxonomy of the burden of treatment and a systematic review of patient-reported measures of burden of treatment in three chronic diseases. The themes covered in the interview with one carer were comparable to the themes discussed by patients. While it would have been desirable to include more carers as participants, our efforts to recruit additional carers as participants during the study period, was unsuccessful. We have added this statement to the discussion section.

5. Finally, the authors do not explain clearly the guide of interview. It is more helpful to have a classical interview guide with three or four questions instead of a semantic map.

Response: The semantic map informed the interview guide but was not the interview guide as such. We have now specified this in the text and added the interview guide as an online supplement.

“The interview guide (see Online Supplement) was derived from our framework of treatment burden in lung cancer, figure …”

4. Concerning the results, it is showed some idea very interesting and the verbatim are really helpful.

Response: We thank the reviewer for the comment. We selected quotes that we felt were representative of the general perspective.

5. The discussion summarises clearly the main results of the study. But, there is lack of analyse with the conceptual framework of the study. Moreover, some elements in connection with the field of health psychology could be seen in the results, in particular ideas relating to the concept of coping or social support that would have been relevant to analyse and put into perspective with this scientific literature.

Response: We have now specified information from the interview analyses was used to update and finalise the treatment burden framework in lung cancer:

“After analysis of all interview data, the preliminary lung cancer treatment burden framework was updated. The final version (figure 1) includes all treatment burden themes that were identified as relevant to lung cancer patients.”

The majority of participants in our study felt well equipped to handle the burden of lung cancer treatment, and we did not explore coping mechanisms in more detail. It would be interesting to examine the concepts of coping and social support in a cohort that includes patients who do not wish to receive treatment or cease treatment prematurely.

Reviewer #2: 

1. This original submission does not provide new or relevant data compared to the literature.

Response: See response to the Editor’s comment (repeated below).

This is the first original study that explores all areas of treatment burden in lung cancer, i.e. the work that patients need to do to manage their condition and the impact that this workload might have on patients’ life. There are a number of studies on treatment burden in other chronic conditions and studies that focused on a single aspect of treatment burden in lung cancer, for example the financial cost or adverse effects of treatment. A cohort study evaluated treatment burden defined as the number of encounters, physicians involved, and medications prescribed in patients with non-small-cell lung cancer [Presley et al, 2017]. In another study, three patients with non-small cell lung cancer were interviewed about how they experienced everyday life during curative radiotherapy [Petri et al, 2015]. There is also a systematic review of qualitative studies that explored the experience of patients with lung cancer or chronic obstructive pulmonary disease and/or their informal caregivers with health or social care interactions [Lippiett at el., 2019]. However, until now, there has been no study that systematically explored all aspects of treatment burden experienced by patients with lung cancer.

We have added this information to the manuscript.

2. Exclusion criteria are not clear: 2 patients still undergoing treatment were excluded (line 133) but patients undergoing treatment could be included (line 94)

Response: We apologise for the confusion. Participants undergoing treatment were included. There were 2 patients undergoing treatment who did not want to participate because they were too busy because of treatment. We have now included them with the other patients who were either too busy or not interested to participate in the study.

3. Figure 1 is illegible

Response: We have reformatted Figure 1, and it should be legible now.

---

## [Editor Report · Decision Letter 1]

22 Dec 2020

PONE-D-20-16636R1

Treatment burden experienced by patients with lung cancer

PLOS ONE

Dear Dr. El-Turk,

Thank you for submitting your manuscript to PLOS ONE. Please check page four and withdraw useless insertion.

We look forward to receiving your revised manuscript.

Kind regards,

Christophe Leroyer

Academic Editor

PLOS ONE

---

## [Author Response · Author response to Decision Letter 1]

25 Dec 2020

Editor: Thank you for submitting your manuscript to PLOS ONE. Please check page four and withdraw useless insertion. 

Response: Thank you for the note, the insertion has been deleted.

---

## [Editor Report · Decision Letter 2]

4 Jan 2021

Treatment burden experienced by patients with lung cancer

PONE-D-20-16636R2

Dear Dr. El-Turk,

We’re pleased to inform you that your manuscript has been judged scientifically suitable for publication and will be formally accepted for publication once it meets all outstanding technical requirements.

Kind regards,

Christophe Leroyer

Academic Editor

PLOS ONE

---

## [Editor Report · Acceptance letter]

12 Jan 2021

PONE-D-20-16636R2 

Treatment burden experienced by patients with lung cancer 

Dear Dr. El-Turk:

I'm pleased to inform you that your manuscript has been deemed suitable for publication in PLOS ONE. Congratulations! Your manuscript is now with our production department. 

Kind regards, 

on behalf of

Dr. Christophe Leroyer 

Academic Editor

PLOS ONE